# The drug development pipeline for glioblastoma—A cross sectional assessment of the FDA Orphan Drug Product designation database

**Pascal Johann**[1,2]*, **Dominic Lenz**[3], **Markus Ries**[3,4]

1 German Cancer Research Center (DKFZ), Division of Paediatric Neurooncology, Heidelberg, Germany,
2 Paediatric and Adolescent Medicine, Swabian Children's Cancer Center Augsburg, Augsburg, Germany,
3 Paediatric Neurology and Metabolic Medicine, Center for Paediatric and Adolescent Medicine, University Hospital Heidelberg, Heidelberg, Germany, 4 Center for Rare Disorders, University Hospital Heidelberg, Heidelberg, Germany

* p.johann@dkfz.de

**Data Availability Statement:** All relevant data are within the manuscript and its Supporting information files.

## Abstract

### Background

Glioblastoma (GBM) is the most common malignant brain tumour among adult patients and represents an almost universally fatal disease. Novel therapies for GBM are being developed under the orphan drug legislation and the knowledge on the molecular makeup of this disease has been increasing rapidly. However, the clinical outcomes in GBM patients with currently available therapies are still dismal. An insight into the current drug development pipeline for GBM is therefore of particular interest.

### Objectives

To provide a quantitative clinical-regulatory insight into the status of FDA orphan drug designations for compounds intended to treat GBM.

### Methods

Quantitative cross-sectional analysis of the U.S. Food and Drug Administration Orphan Drug Product database between 1983 and 2020. STROBE criteria were respected.

### Results

Four orphan drugs out of 161 (2,4%) orphan drug designations were approved for the treatment for GBM by the FDA between 1983 and 2020. Fourteen orphan drug designations were subsequently withdrawn for unknown reasons. The number of orphan drug designations per year shows a growing trend. In the last decade, the therapeutic mechanism of action of designated compounds intended to treat glioblastoma shifted from cytotoxic drugs (median year of designation 2008) to immunotherapeutic approaches and small molecules

**Funding:** The author(s) received no specific funding for this work.

**Competing interests:** The authors have declared that no competing interests exist.

**Abbreviations:** ALL, Acute lymphoblastic leukemia; AML, Acute myeloid leukemia; CAR, Chimeric antigen receptor; CLL, Chronic lymphocytic leukemia; EGFR, Epidermal growth factor receptor; FDA, Food and drug administration; GBM, Glioblastoma; GIST, Gastrointestinal stroma tumour; HCC, Hepatocellular carcinoma; HL, Hemophagocytic lymphhistiocytosis; HSV, Herpes simplex virus; MB, Medulloblastoma; MDS, Myelodysplastic syndrome; MGMT, Methylguaninmethyltransferase; MPN, Myeloproliferative neoplasia; NC, Nasopharyngeal carcinoma; NTRK, Neurotrophic tyrosine kinase; SCD, Sickle cell disease; VEGF, Vascular endothelial growth factor; VEGFR, Vascular endothelial growth factor receptor.

(median year of designation 2014 and 2015 respectively) suggesting an increased focus on precision in the therapeutic mechanism of action for compounds the development pipeline.

## Conclusion

Despite the fact that current pharmacological treatment options in GBM are sparse, the drug development pipeline is steadily growing. In particular, the surge of designated immunotherapies detected in the last years raises the hope that elaborate combination possibilities between classical therapeutic backbones (radiotherapy and chemotherapy) and novel, currently experimental therapeutics may help to provide better therapies for this deadly disease in the future.

## Introduction

High grade gliomas account for the majority of brain tumour related deaths in children and adults. Considering all age groups together, glioblastoma represents the most common malignant brain tumour (43,5% of all malignant brain tumours [1]).

Albeit being rare in absolute numbers, glioblastomas represent a universally fatal disease class for approximately 15,000 patients per year in the United States [1]. While the last years have seen a surge in publications that highlight intertumoural and intratumoural diversity [2, 3] in these tumours, our growing understanding of the pathophysiological processes that underlie the disease could so far not yet be translated into therapeutic success.

To date, a wealth of studies has identified the typical genetic alterations that occur in glioblastoma: Mutations in *IDH1* or, for paediatric glioblastomas, the two frequently occurring histone *H3.3* gene mutations (*H3.3*: pK27M and *H3.3*: pG34R/V) are just three examples of common genetic mutations that define distinct molecular classes of glioblastoma. The well-known mutations identified in glioblastoma have subsequently lead to the identification of epigenetic and transcriptomic mechanisms which perpetuate the disease: examples of this are the hypermethylation of CpG islands in IDH1-mutant glioblastoma [4] and the loss of histone H3.3 K27me3 in H3.3 mutant glioblastomas [5].

Despite the vast increase in knowledge about the genome, epigenome and transcriptome of glioblastoma, clinical outcomes have not changed and drug development in glioblastoma is lagging behind the significant advances in glioblastoma (epi)genomics. While some of these genetic targets can be used therapeutically, the majority of them are unsuitable as a drug targets although they may offer the prospect of use in immunotherapy.

Thus, there is an unequivocal medical need for novel compounds or combinations of compounds that are able to put a hold on disease progression.

The U.S. Orphan Drug Act of 1983 was intended to incentivize drug development in rare diseases including rare cancers by providing protocol assistance, orphan grants programs, tax credit for 50% of clinical trial costs, U.S. Food and Drug Administration (FDA) fee waiver, and 7 years of marketing exclusivity [6]. Between 1983 and 2015, more than a third of all orphan drug approvals (N = 177 out of a total of 492, i.e., 36%) were related compounds intended to treat rare cancers [7].

While there may be manifold reasons for a clinical failure of novel drugs, a comprehensive view on the status of designated compounds for the indication glioblastoma is still lacking.

In particular, it remains unclear which substance classes and therapeutic principles for glioblastoma have entered the market or are under development. This knowledge is instructive as

the pharmacological principles which underlie the designated drugs may have changed over time and thus may mirror the different directions of brain tumour research. We aim to analyse the lessons that we have learned by assessing successes and failures in orphan drug development in glioblastoma. Therefore, we present a cross-sectional, quantitative clinical-regulatory insight into the status of FDA orphan drug designations for compounds intended to treat GBM. This study covers the period between January 1983 and August 2020.

## Methods

STROBE criteria (S1 Checklist) were respected for planning, conduct, analysis, and reporting of this study [8]. We accessed the Orphan Drug Product designation database on 30 July 2020 at https://www.accessdata.fda.gov/scripts/opdlisting/oopd/ and downloaded the information on all designated drugs using the search term "Glioblastoma". The list of drugs was then manually cleared from non-oncological indications. An allocation to the field of "paediatric oncology" or "adult oncology" was performed by a board-certified paediatric oncologist.

Disease entities which typically occur both in adult age and in the field of paediatric oncology (such as lymphomas and osteosarcoma for instance) were allocated to both categories. Others which almost typically occur in paediatric oncology such as ALL were considered only for this area.

Subsequently, designated drugs that were intended to treat glioblastoma were characterized according to their mode of action in the pharmacological classes "cytostatics", "cellular/viral immunotherapy", "targeted therapies" or "others". Targeted therapies were defined as substances for which at least one molecular target could be identified by literature research [9]. Compounds that could not be classified unequivocally were categorized as "others"–this class also contained compounds that are being used as diagnostics.

In order to independently verify whether there were approved drugs for the treatment of glioblastoma that were not listed in the U.S. Food and Drug Administration Orphan Drug Product database, we conducted a full text search in the FDA drug label database (FDALabel, https://nctr-crs.fda.gov/fdalabel/ui/search). Search terms were "glioblastoma" in the section "indications and usage". The database was accessed on 28 October 2020. Findings were juxtaposed to the approved compounds identified from the search in Orphan Drug Product designation database as described above. In addition, we cross-validated whether or not the compounds identified from the search in FDALabel were registered as orphan drugs.

Standard methods of descriptive statistics were applied. In particular, continuous variables were summarized with mean, standard deviation, and median, minimum, and maximum values whereas categorical variables were summarized with frequencies and percentages. Analyzed groups included 1) approved and 2) designated compounds intended to treat glioblastoma.

In order to determine the number of approved drugs, we first queried the downloaded data from the Food and Drug Administration Orphan Drug Product database for FDA approved drugs and curated the list by eliminating duplicate terms. Likewise, the data were analyzed for designated compounds. In addition, we analyzed number and characteristics of orphan drug designations for glioblastoma that were subsequently withdrawn. In order to put our findings on orphan drug designations for glioblastoma into perspective within a global oncological context we analysed orphan drug designations for all oncological indications currently listed in the US Food and Drug Administration Orphan Drug Product database. For the review of compounds which have been used in Glioblastoma trials, we accessed the clintrials.gov database (URL: https://clinicaltrials.gov/) on 07th of April 2021 and downloaded all interventional trials that were completed for the search term "glioblastoma" and for which data have been

published. The compounds that were used in these trials were classified into the same categories that were applied for the designated compounds.

For statistical analysis and graphical display, we used the software R (version 3.5.0). For plots, the program library ggplot2 was employed. We used the CONSORT checklist when writing our report [10]

Patient and public involvement: No patient involvement.

## Results

### Approved drugs for glioblastoma

A total of four compounds (Table 1) were approved by the FDA for the indication glioblastoma. Three of them were therapeutic compounds, one is 5-aminolevulinic acid which is a photo-diagnostic substance for the intraoperative detection of resection margins [11].

These results were in line with the findings of the FDALabel database query: bevacizumab, carmustine, and temozolomide list the indication "glioblastoma" on their FDA approved labels. All of these drugs have been used in the clinical setting, however with discouraging results and no improvement in the outcome of glioblastoma [12].

The small number of approved compounds prompted us to further explore the drug developmental landscape in this disease.

### Designated drugs in glioblastoma

Given the low number of approved drugs that can be used in the clinical setting, we next sought to get an overview on the drug development landscape in glioblastoma, trying to quantify and qualify the drugs that are in the pipeline for this indication.

Overall, 162 compounds had an orphan drug designation for glioblastoma. Until 2016, the number of drugs designated for the indication glioblastoma varied, but on average displayed an increasing trend (Fig 1). For the last four years, this tendency seems to have reversed and the number of designated compounds per year displays a downward trend. To be able to put these findings in GBM into a global oncological drug development context, we assessed the spectrum of all oncological—and paediatric oncological diseases with orphan drug designations.

Our analysis here yielded 4618 compounds (out of a total of 5513 orphan drug designations for any rare disease = 83%, Fig 2) that were designated for either a paediatric or an adult oncological entity. As in glioblastoma, since 1984, the number of orphan drug designations for adult oncology per year varied with an increasing trend and showed a peak of 342 compounds in 2016. As expected, the number of compounds targeting typical disease entities from adult oncology was consistently higher than in paediatric oncology (on average 37% higher). The numeric pattern over time appeared to be similar in paediatric and adult orphan drug designations.

**Table 1. An overview on approved compounds for the indication "glioblastoma".**

| Name | Target structure | Year of Approval | Year of designation |
|---|---|---|---|
| 5-aminolevulinic acid | intraoperative optical imaging agent | 2017 | 2013 |
| Bevacizumab | Inhibition of angiogenesis (antibody against VEGF) | 2009 | 2006 |
| Polifeprosan 20 with carmustine | Implant to deliver the approved drug carmustine | 2003 | 1989 |
| Lomustine | Alkylating compound | 1976 | unknown |
| Temozolomide | Alkylating compound | 2006 | 1998 |

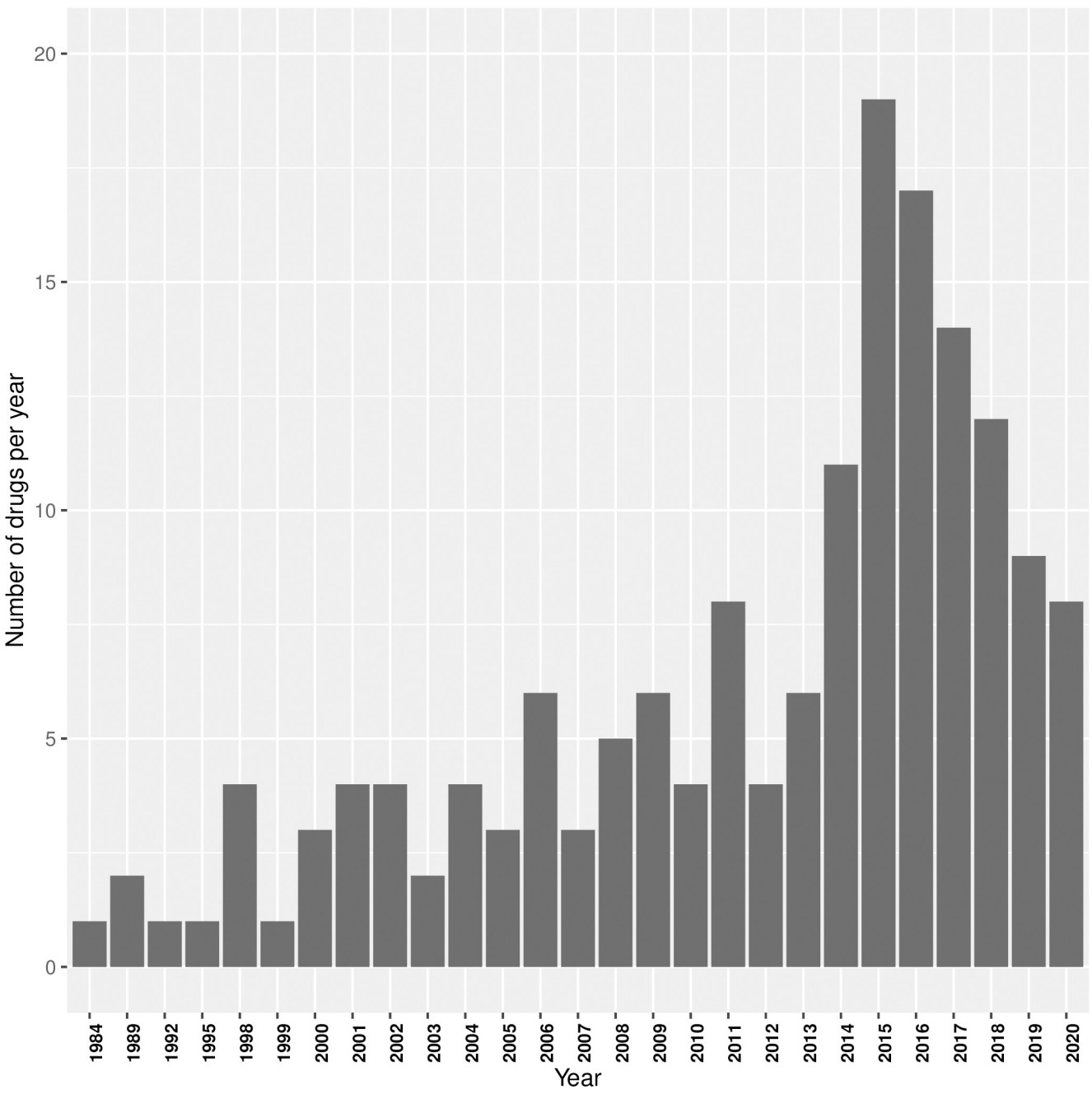

**Fig 1. Barplot shows the number of new orphan drug designations for the indication glioblastoma per year (years without a new designation are not shown).**

To better understand the intended indications of these designated compounds, we then analyzed, which tumour entities are targeted by these drugs.

Fig 3 shows the frequency distribution of FDA orphan drug designations for their respective oncological indications. Most oncological orphan drug designations for the adult patient population were granted for lymphoma, pancreatic cancer, and glioblastoma. In contrast,

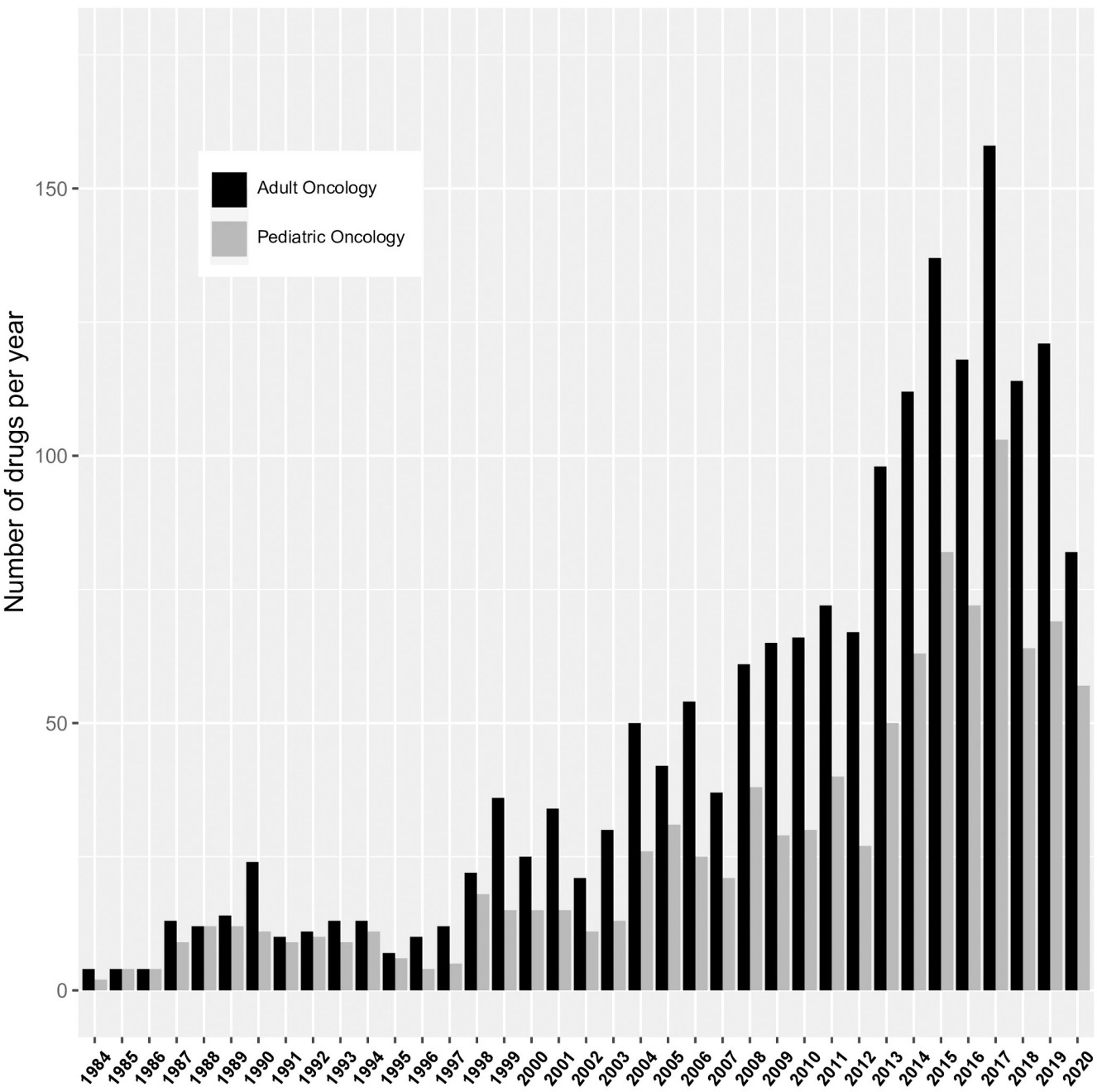

**Fig 2. Barplot shows the number of orphan drug designations in paediatric oncology and in adult oncology per year.**

lymphomas, glioblastoma and AML received the majority of orphan drug designations for paediatric cancers.

(Fig 3A and 3B). Although some of these categories are quite heterogeneous and comprise different entities (such as lymphomas which are in fact a group of genetically heterogeneous diseases associated with divergent outcome), the predominance of these groups is remarkable as they do not represent the oncological indications which occur most frequently but which are associated with a high mortality. Thus, the designated compounds in fact address the

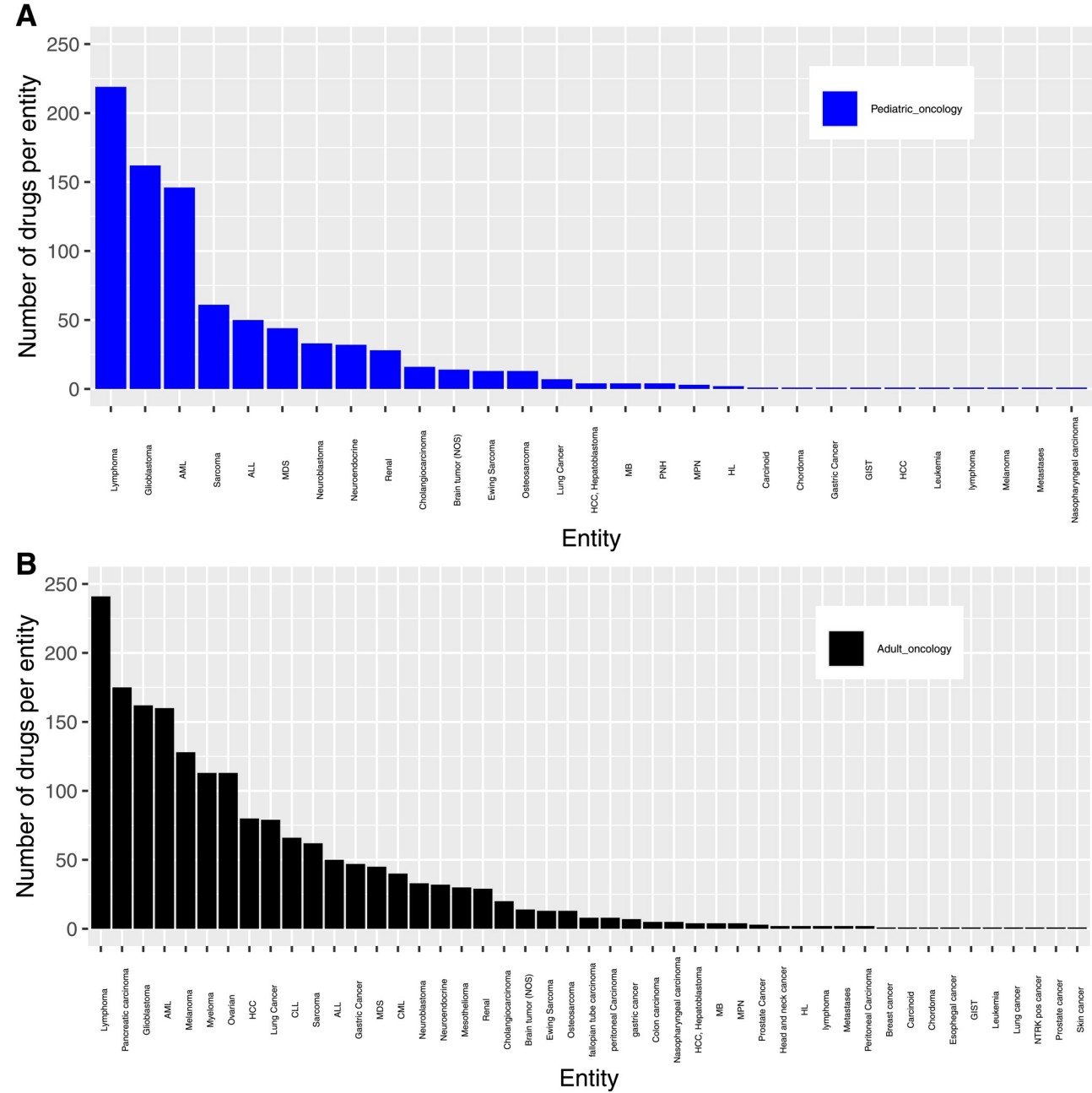

**Fig 3. Barplot shows the distribution of entities in paediatric (A) and adult (B) oncological entities with orphan drug designations.**

unmet medical need of cancers which are associated with a high mortality despite not being the most frequent ones [13].

## Withdrawn orphan drug designations in glioblastoma (S1 Table)

Studying the compounds designated for glioblastoma which were subsequently withdrawn from the market is instructive as it may highlight potential mechanistically interesting substances that never reached the clinic.

Our search in the FDA approved drugs database revealed 14 designated compounds which were subsequently withdrawn from the drug development pipeline. Some of these drugs were classical cytostatics, others displayed more innovative modes of action: cilengitide, for instance, an integrin inhibitor [14] was among the withdrawn substances. Other, less well-known substances included the glutamate receptor inhibitor talampanel and cintredekin besudotox, an IL13 conjugated toxin, specifically targeting glioblastoma. Reasons for these withdrawals were not published and are therefore, unfortunately, not known.

## Pharmacological classes of designated drugs in glioblastoma

We next characterized the pharmacological classes which were designated per year. We therefore allocated the designated compounds into the broad categories "cytostatics", "targeted therapy", "Cellular product/ Virus" and "others". The latter constitutes a heterogeneous group of substances comprising intraoperative fluroescent dyes (as diagnostics), peptide vaccinations or repurposed drugs such as cannabinoids which are approved for other indications and were subsequently found to display anti-neoplastic properties ("repurposing").

When regarding the designation per compound class over time, we found that the median year of designation for cytostatic drugs was 2008 (Fig 4A). With an increasing knowledge on both the genetic makeup of glioblastomas and the resulting therapeutic targets, the years 2010–2020 saw an increase in small molecule inhibitors, directed against specific molecular structures (Fig 4A). An investigation on the nature of these therapeutic targets revealed a high diversity: Compounds directed against VEGF or the VEGFR were most frequently found, but there were also molecules directed against EGFR—a molecule prototypically mutated in subsets of adult glioblastoma [2].

Similarly, immunotherapeutic approaches including dendritic cell vaccinations, or NK cell/ T-cell based therapies represents a focus of the last years compound designations. As cellular and viral therapies represent a very diverse group, we dissected this category further (Fig 4B): Remarkably, 45% (9/20) of all therapeutics proved to be virus based, the majority of which being oncolytic viruses. In dendritic cell based therapies, the second largest group, the dendritic cells were mostly stimulated with autologous tumour lysates or with synthetic peptides derived from glioblastomas, aiming to elicit an anti-tumour immune response in the host. The glioma-based therapeutics mainly consist of autologous tumour cells, which were engineered to express immunogenic peptides/antigenes (such as an aberrant IGF1-R receptor).

Although none of the latter therapeutics has been approved for glioblastoma so far, the number of designated drugs in this category points to a high potential of these compounds in the clinic.

To examine which designated drugs have been used in recently in the frame of completed and ongoing clinical studies, we classified compounds that were contained in interventional studies from the portal clintrials.gov (S2 Table). Among the completed drug trials, the majority (102/141; 72,3%) included at least one compound designated or approved for the indication "glioblastoma". Notably many of these studies combined an approved compound (such as bevacizumab or temozolomide) with a more experimental drug. Among the completed studies, only very few (5/141, 3,54%) made use of immunotherapeutic approaches either alone or in combination with cytostatics, a number that may possibly rise in the future.

## Discussion

### Approved drugs in glioblastoma

The overall increase in orphan drug designations for the whole oncological field is also seen in the case of glioblastoma (with an average of six designated drugs per year). In stark contrast to

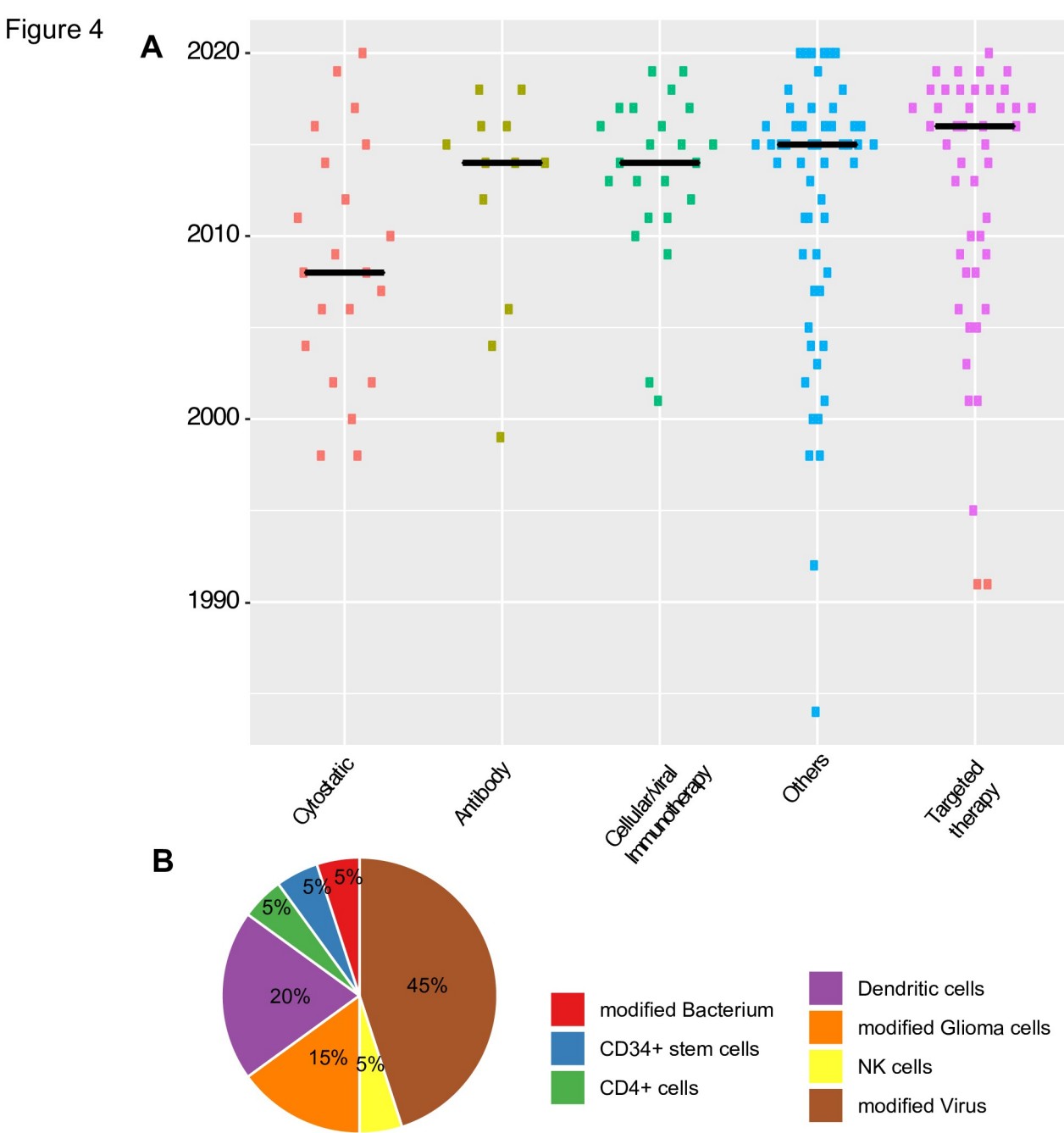

**Fig 4.** A) Dotplots show the substance classes of designated drugs in GBM, B) Pie chart shows the mode of action of designated immunotherapies/ cellular therapies for glioblastomas.

the number of designations, only six compounds were approved for this entity in the last 30 years—the most recent substance being bevacizumab, an antibody that targets VEGF, which however did not show a survival benefit in large, placebo-controlled studies [15]. Thus, considerable discussions are ongoing about whether the FDA-approval of bevacizumab should be withdrawn again.

Other approved compounds for GBM include cytostatic drugs such as carmustine or temozolomide. Temozolomide has become a frequently used standard therapy due to its generally favourable toxicity profile. It is one of few drugs for which a biomarker has been identified: The MGMT promoter governs expression of the corresponding gene. It represents the most important resistance mechanism to an alkylating therapy and its hypermethylation has been associated with a better outcome [16].

It is remarkable that so far no cellular or virus based immunotherapy has been granted approval by the FDA, although there have been promising preclinical and clinical [17] studies suggesting a potential benefit.

Of particular during the last years, the number of designated cellular therapies has increased. Some of them employ T-cells with chimeric antigen receptors, others back on dendritic cell vaccinations.

## Spectrum of indications for designated drugs

Overall, our study revealed a wealth of compounds designated for oncological indications and an average increase in drug designations per year over time. However, this trend seemed to continue only until the year 2016 which marked a turning point with a decrease in designated compounds per year from then on. Reasons for this decline in the last years may be manifold. While the Corona-pandemic may have influenced the number of designated drugs in 2020, the reasons for the receding numbers in the years 2017–2019 could be a lack of novel anti-neoplastic agents that successfully undergo testing in clinical studies. Other reasons that could negatively influence the process of compound designation are a lack of funding, changes in the regulatory environment, or unsustainable businesses which may have grown in these years.

While some of the substances that we highlight here were already discussed and contained in a review by Lassen et al. [18], the surge in immunotherapeutic approaches which we highlight here seems noteworthy. For many of these novel, immunotherapeutic products, no efficacy data in the sense of randomized, placebo-controlled trials have been published. However, safety of administration and first-in-human data are available for a number of medications: The oncoloytic HSV-1 (G207) has demonstrated safety in a phase I study with a median survival of 7,5 months (after inoculation of the virus) [19] and good tolerability of the modified virus. For another virus based immunotherapy (employing the genetically modified HSV M032), a clinical protocol and non-human primate data on safety have been published, however Phase I data are not available to date [20].

Similar data exist for other immunotherapeutic products that use modified progenitor cells: Hematopoietic progenitor cells, transfected with a mutant MGMT-receptor to enhance temozolomide resistance of the hematopoietic system, were used within the frame of a phase I and demonstrated a reasonable safety profile [21].

Overall, none of these medications has demonstrated ground-breaking progress in overall survival in these preliminary data. However, promising Phase 1 data exists which is worthwhile to be pursued further.

It is remarkable, that the majority of designated compounds in adult oncology targeted lymphomas, pancreatic cancer and glioblastoma. This does not necessarily reflect the epidemiological spectrum of malignant diseases with breast cancer, lung cancer and prostate cancer being the most frequent cancers. When, however, considering cancer mortality from these entities, the designations certainly do meet a medical need.

## Withdrawn drugs in glioblastoma

Several orphan drug designations for glioblastoma were subsequently withdrawn without ever having been approved. Unfortunately, the precise reason for these withdrawals is not known. It would be interesting to capture this information transparently in public or even in the clinicaltrials.gov database as this would allow the scientific community to learn from previous experiences, and potentially avoid unnecessary exposure of subjects to clinical research. Possible reasons for failure may include a flawed scientific rationale, flawed trial design or unsustainable business (https://termeerfoundation.org/collaborations accessed 06 October 2020 [22]).

## Drug safety considerations and innovative aspects of drug development in glioblastoma

Usually orphan drug development programs involve fewer patients and fewer clinical trials than non-orphan drug development programs. No approved drug for glioblastoma was withdrawn. This indicates that there were no major safety issues in the orphan drug development process in this area that were detected in the post-approval pharmacovigilance process. With respect to innovation in the process of granting approval to novel drugs, there is certainly room to expedite this process: The only targeted drug among the approved compounds remains bevacizumab. Until today, the impact of the US orphan drug act on the drug development for glioblastoma has been limited: There are only four FDA orphan approvals for the treatment of glioblastoma—one of them (5-aminolevulinic acid.) is a diagnostic compound. There is, however, hope for progress. It is possible that more compounds may successfully reach the clinic, because 60 orphan drug designations have been granted with an increasing tendency in the last 5 years.

## Barriers to a successful translation of preclinical findings to the clinic

GBM represents a genetically highly complex disease: When progressing, these tumours undergo a complex, molecular evolution that results in an increase of genetic aberrations. Thus, therapies which target only one specific molecular lesion fall short in controlling the diverse number of other pathways which may contribute to tumour growth. However even combination therapies (including small molecule inhibitors and classical cytotoxic compounds) did not show the desired effects in clinical studies).

Further problems which are encountered in the management of glioblastoma include its invasive nature and the impossibility to achieve a gross total resection owing to the infiltrative growth, its high proliferation rate and the associated speed by which resistance mechanisms toward applied therapies emerge. Despite a growing number of designated compounds and molecular directed therapeutics only few of them address the molecular characteristics of the genetic and epigenetic glioblastoma subgroups. As an example, the molecular mechanism that drives H3K27M mutant glioblastomas is well described by now: A sequestration of the enzyme PRC2 [5] leads to a global loss of the repressive histone mark H3K27me3. An inhibitor of enzymes (GSK-J4) that catalyze the demethylation of H3K27me3, thus restoring this mark, has been tested [23] but so far has not reached the status as a designated drug.

There is hope that immunotherapy- either antibody based on the inhibition of PD(L)1 or by using cellular products such as CAR-T cells [24] or dendritic cells—may ultimately improve the outcome for patients with glioblastoma. At the least, these agents represent combination partners that may expand the portfolio of classically used cytotoxic drugs.

## Limitations

This analysis of orphan drug development in the field of oncology is limited to the data provided in U.S. Food and Drug Administration Orphan Drug Product database. Other geographic regions were not included in this study because the FDA database is considered comprehensive. As orphan drug development is generally a global endeavour, we consider the results of this analysis to be generalizable within the context of these limitations.

In summary, we conclude despite the fact that current pharmacological treatment options in GBM are sparse, the drug development pipeline in GBM has been growing steadily until 2016 and the number of designated drugs is still at a high level since then [25]. In particular, the surge of designated immunotherapies during the last years raises the hope that elaborate combination therapies between classical therapeutic backbones (i.e. radiotherapy and chemotherapy) and these novel, currently experimental interventions may help to provide better treatment options for this deadly disease in the future.

## Supporting information

**S1 Checklist. Overview on the STROBE criteria which were respected for the data analysis.**
(DOC)

**S1 Table. Table on withdrawn drugs in glioblastoma and description of molecular targets where available.**
(XLSX)

**S2 Table. An overview on completed drug trials in glioblastoma with compounds that were used in these studies and a classification of compounds.**
(XLS)

## Acknowledgments

We kindly thank Dr. Hanna Seidling (Cooperation Unit Clinical Pharmacology, Heidelberg) for help with the classification of designated compounds. We thank Lorna Stimson, PhD, for language editing.

## Author Contributions

**Conceptualization:** Pascal Johann, Markus Ries.

**Data curation:** Pascal Johann, Dominic Lenz.

**Formal analysis:** Pascal Johann, Dominic Lenz, Markus Ries.

**Investigation:** Pascal Johann, Markus Ries.

**Methodology:** Pascal Johann.

**Writing – original draft:** Pascal Johann, Markus Ries.

**Writing – review & editing:** Dominic Lenz.

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
