## [Decision Letter · Decision Letter 0]

29 Mar 2021

PONE-D-21-03939

The drug development pipeline for glioblastoma - a cross sectional assessment of the FDA Orphan Drug Product designation database

PLOS ONE

Dear Dr. Johann,

Thank you for submitting your manuscript to PLOS ONE. After careful consideration, we feel that it has merit but does not fully meet PLOS ONE’s publication criteria as it currently stands. Therefore, we invite you to submit a revised version of the manuscript that addresses the points raised during the review process.

The essential points that require addressing are as follows:

Provide a Supplementary Table summarizing drug trials in both pediatric and adult GBM use for analysis, including all vaccines in the same cellular product/virus category, and renaming that category appropriately (see reviewers 1 & 4 remarks pertaining to Figure 4A, below).

Correct all errors and clarify all points (including captions) raised by each reviewer in Table and Figures.

Revise the main conclusion that the number of orphan drugs being developed is escalating as per reviewer 3's comment on Figure 1, which shows a peak in 2016 and a downward trend since.

There were no significant conflicts between reviewers. Addressing all other points raised by them is therefore also strongly recommended.

I believe that your manuscript represents an informative summary of the US orphan drug development program for glioblastoma. If appropriately revised, it will be particularly useful for researchers working on new drug development, as well as for clinicians treating patients afflicted with this devastating tumor.

We look forward to receiving your revised manuscript.

Kind regards,

Christopher Wheeler, Ph.D.

Academic Editor

PLOS ONE

Journal Requirements:

2. To comply with PLOS ONE submission guidelines, in your Methods section, please provide additional information regarding your statistical analyses. For more information on PLOS ONE's expectations for statistical reporting, please see https://journals.plos.org/plosone/s/submission-guidelines.#loc-statistical-reporting.

3. Please ensure that you refer to Figure 1 in your text as, if accepted, production will need this reference to link the reader to the figure.

Reviewers' comments:

Reviewer's Responses to Questions

**Comments to the Author**

1. Is the manuscript technically sound, and do the data support the conclusions?

Reviewer #1: Yes

Reviewer #2: Yes

Reviewer #3: Partly

Reviewer #4: Partly

2. Has the statistical analysis been performed appropriately and rigorously? 

Reviewer #1: Yes

Reviewer #2: N/A

Reviewer #3: Yes

Reviewer #4: N/A

3. Have the authors made all data underlying the findings in their manuscript fully available?

Reviewer #1: Yes

Reviewer #2: No

Reviewer #3: Yes

Reviewer #4: Yes

4. Is the manuscript presented in an intelligible fashion and written in standard English?

Reviewer #1: Yes

Reviewer #2: No

Reviewer #3: No

Reviewer #4: Yes

5. Review Comments to the Author

Reviewer #1: The authors relay information obtained from a searchable FDA orphan drug database of compounds approved or being investigated for the treatment of glioblastoma brain tumor.

The authors identified four compounds approved for use in treatment or diagnosis or glioblastoma. 162 additional compounds had an orphan drug designation not FDA approved for glioblastoma. The authors also found a steady increase of orphan drug designations over time from 2016.

Mention is made of the main categories of investigated compounds for glioblastoma. The manuscript would benefit from more in-depth analysis of the promising compounds and mechanisms of action to understand the main directions that investigators are taking in search of a more successful treatment approach to glioblastoma. .

Reviewer #2: This article provides an informative summary of the status of FDA orphan drug designations GBM. However, there are many errors both grammatical and content that need to be addressed before publication.

Please clarify whether reporting for GBM only or high-grade glioma.

Provide Supplementary Table summarizing drug trials in both pediatric and adult GBM use for analysis, and list designated categories (cytostatic, antibody, cellular/virus, targeted therapy, other)

Suggest including all vaccines in the same category, peptide and dendritic cell. It would seem most appropriate to include in the cellular/virus category and to rename appropriately.

Table 1: There are many errors in this table, including 1) Content errors, 2) misspellings “ame”, “copound”. Consider editing “Target Structure” to “Description”. Also, it is not clear what year of designation refers to. Is this the year approved by the FDA or year of orphan drug designation? Please clarify and check all dates. Further, there are 5 approved drugs for GBM (not 4): 5-aminolevulinic, bevacizumab, temozolomide, lomustine and carmustine.

- 5-aminolevulinic: better described as an “intraoperative optical imaging agent” vs “diagnostic” or “fluorescent dye”. Also, it received ODD by the FDA in , and FDA approval in 2017. It is not clear what specific regulatory event is being listed as 2002.

- Bevacizumab: This is an anti-VEGF antibody not a radioconjugate small molecule. Also, bev was approved by the FDA in 2009 so again it is not clear the regulatory event being listed as 2014.

- Prolifeprosan 20 with carmustine (GLIADEL) is an implant to deliver the approved drug carmustine. This needs to be clarified.

- Temozolomide: check year of designation date.

Overuse of “:”. In most contexts it would be more appropriate to replace the “:” and instead start a new sentence. If the authors choose to use a “:” to join 2 related sentences, then the second sentence should not be capitalized.

Review should be carefully proof-read.

Reviewer #3: The authors present a cross sectional assessment of the US orphan drug development program with regards to glioblastoma. This original research provides a helpful overview of the evolving treatment landscape for GBM, even while highlighting the relative paucity of approved therapies thus far.

The statistical analysis in this article is largely descriptive and not problematic. However, the authors' main conclusion that the number of orphan drugs being developed is escalating is not supported by Figure 1, which shows a peak in 2016 and a downward trend since. (2020 results could be affected by COVID-19, but there isn't a clear explanation for 2017-2019.) This doesn't invalidate the analysis, but it should modify the conclusion.

Other comments:

- The official WHO term is now "glioblastoma" not glioblastoma multiforme.

- Clinical outcomes for glioblastoma HAVE improved, just not much.

- Table 1 is not accurate. Bevacizumab is the angiogenesis inhibitor, BCNU wafers are not.

- Might be worth noting in the discussion that of these approved therapies, usage of 5-ALA and BCNU wafers is not universal, thus further highlighting the need for more progress.

- Characterizing cilengitide as "well known" and other drugs as "less well-known" seems somewhat arbitrary and akin to a personal opinion.

- "Cannaboids" is misspelled.

- "Temozolomide" is misspelled.

- I don't think it's accurate to "assume" that CAR T-cells or dendritic cell vaccines will ultimately be approved for GBM. There is no evidence to support this at this time.

- Might be worth discussing that although bevacizumab is still FDA-approved for recurrent GBM, there was considerable discussion about pulling this approval when phase 3 trials demonstrated the absence of a strong survival benefit.

- The GSK-J4 example doesn't seem to fit this article, as it is not a orphan designated drug.

Reviewer #4: This manuscript gives a comprehensive overview of the landscape of designated and approved orphan drugs to treat glioblastoma multiforme. The information provided is useful for researchers who work on developing new drugs for GBM or clinicians interested in approved drugs and recent trends. However, the paper has several flaws that need to be corrected (see below).

The conclusion that the rate of designated (but not yet approved) drugs per year has steadily increased is not supported by figures 1 and 2, which show a rapid rise in new designated drugs starting about 10 years ago, reaching a maximum in 2016 and then declining to the level of the beginning of the decade. The authors should discuss this trend and potential reasons for the reversal in the Discussion section.

A similar review, entitled "Orphan drugs in glioblastoma multiforme: a review," was published by Lassen et al. in 2014 (https://doi.org/10.2147/ODRR.S46018). The authors should reference that paper and make it clear that their review adds new information.

Table 1 has many errors, including typos (ame, copound) and wrong drug information under Target structure and Year of Designation. Please correct these errors.

The caption of Figure 1 should be "Barplot shows the number of new orphan drug designations for the indication glioblastoma per year. Years without a new designation are not shown.

The vertical axis label of Figure 1 should be "Number of new drugs per year." Make the same changes to FIgure 2.

In Figure 3A, it seems that the Lymphoma category does not have a bar.

In Figure 3A and B, the authors included non-oncological, rare diseases such as Thalassemia and SCD (Sickle Cell Disease). They should carefully check each category and remove all non-oncological classifications.

In Figure 4 A, I suggest replacing 'Cellular product/Virus' with 'Cellular/Viral Immunotherapy' for clarity.

In Figure 4 A, the individual plots are 'dot plots' not 'boxplots'; no need to capitalize that phrase.

In the Abbreviations table, the entries PNH Paroxysmal nightly hemoglobinuria and SCD Sickle cell disease should be removed because they are rare genetic diseases; also see comment about Figure 3 A and B above.

In the Abbreviations table, replace 'Vasoendothelial growth factor' with 'Vascular endothelial growth factor'.

In the Abbreviations table, add VEGFR - Vascular endothelial growth factor receptor.

6. PLOS authors have the option to publish the peer review history of their article (what does this mean?). If published, this will include your full peer review and any attached files.

Reviewer #1: No

Reviewer #2: No

Reviewer #3: No

Reviewer #4: No

---

## [Author Response · Author response to Decision Letter 0]

22 Apr 2021

PONE-D-21-03939

The drug development pipeline for glioblastoma - a cross sectional assessment of the FDA Orphan Drug Product designation database

PLOS ONE

We thank the editor and the reviewers for their thoughtful comments that were very helpful to further strengthen our manuscript.

We took all the issues raised into account and address each of them point-by-point in the following section: 

Editor

Comment 1: Provide a Supplementary Table summarizing drug trials in both pediatric and adult GBM use for analysis, including all vaccines in the same cellular product/virus category, and renaming that category appropriately (see reviewers 1 & 4 remarks pertaining to Figure 4A, below).

Correct all errors and clarify all points (including captions) raised by each reviewer in Table and Figures.

Answer 1: This has now been changed: We have generated a supplementary Table (Supplementary Table 3) that gives an overview on completed drug trials in GBM and classifies the compounds that are used in these interventional studies. We have inserted the category “cellular/viral immunotherapy” for that purpose. 

Comment 2: Revise the main conclusion that the number of orphan drugs being developed is escalating as per reviewer 3's comment on Figure 1, which shows a peak in 2016 and a downward trend since.

Answer 2: We have now corrected this statement (p. 9: “Until 2016, the number of drugs designated for the indication glioblastoma varied, but displayed an increasing trend (Figure 1). (…)””. Moreover, we have inserted a reference pointing to Figure 1 in the text.

Comment 3: There were no significant conflicts between reviewers. Addressing all other points raised by them is therefore also strongly recommended. I believe that your manuscript represents an informative summary of the US orphan drug development program for glioblastoma. If appropriately revised, it will be particularly useful for researchers working on new drug development, as well as for clinicians treating patients afflicted with this devastating tumor.

Answer 3: Thank you very much for this encouraging comment. We are addressing all points raised by the reviewers.

Journal Requirements:

Comment 1: Please ensure that your manuscript meets PLOS ONE's style requirements, including those for file naming. The PLOS ONE style templates can be found at

Answer 1: This has now been taken care of, the style requirements have been respected for this revision.

Comment 2: To comply with PLOS ONE submission guidelines, in your Methods section, please provide additional information regarding your statistical analyses. For more information on PLOS ONE's expectations for statistical reporting, please see https://journals.plos.org/plosone/s/submission-guidelines.#loc-statistical-reporting.

Answer 2: This has now been taken care of.

Comment 3: Please ensure that you refer to Figure 1 in your text as, if accepted, production will need this reference to link the reader to the figure.

Answer 3: We have now inserted a reference, pointing to Figure 1 in the manuscript text

Comment 4: Please include captions for your Supporting Information files at the end of your manuscript, and update any in-text citations to match accordingly. Please see our Supporting Information guidelines for more information: http://journals.plos.org/plosone/s/supporting-information.

Answer 4: Thank you very much for this comment, this is now also implemented at the end of the manuscript

Reviewer #1: 

Comment 1: The authors relay information obtained from a searchable FDA orphan drug database of compounds approved or being investigated for the treatment of glioblastoma brain tumor.

The authors identified four compounds approved for use in treatment or diagnosis or glioblastoma. 162 additional compounds had an orphan drug designation not FDA approved for glioblastoma. The authors also found a steady increase of orphan drug designations over time from 2016.

Mention is made of the main categories of investigated compounds for glioblastoma. The manuscript would benefit from more in-depth analysis of the promising compounds and mechanisms of action to understand the main directions that investigators are taking in search of a more successful treatment approach to glioblastoma. 

Answer 1: We are very grateful to the reviewer for the positive overall feed-back of our work. To present a more detailed discussion of promising, recently designated compounds, we have now improved the discussion section by providing a more detailed analysis on some of the designated compounds and compound categories (section “Spectrum of indications for designated drugs”). In particular, we focus on the data of the immunotherapeutic approaches which are currently being pursued. 

Reviewer #2

Comment 1: This article provides an informative summary of the status of FDA orphan drug designations GBM. However, there are many errors both grammatical and content that need to be addressed before publication. Please clarify whether reporting for GBM only or high-grade glioma.

Answer 1: We thank the reviewer for this important remark. The analysis is to glioblastoma although many of the compounds presented here may be applicable to other high grade gliomas as well. We have clarified this in the "Methods" section of the manuscript.

Comment 2: Provide Supplementary Table summarizing drug trials in both pediatric and adult GBM use for analysis, and list designated categories (cytostatic, antibody, cellular/virus, targeted therapy, other)

Suggest including all vaccines in the same category, peptide and dendritic cell. It would seem most appropriate to include in the cellular/virus category and to rename appropriately.

Answer 2: We have now made the required changes and summarized all vaccines in one category. Moreover, we have generated a table (Supplementary Table 3) based on the clintrials.gov database that lists completed and ongoing trials in GBM and highlights whether any of the FDA-designated compounds are used in these.

Comment 3: Table 1: There are many errors in this table, including 1) Content errors, 2) misspellings “ame”, “copound”. Consider editing “Target Structure” to “Description”. Also, it is not clear what year of designation refers to. Is this the year approved by the FDA or year of orphan drug designation? Please clarify and check all dates. Further, there are 5 approved drugs for GBM (not 4): 5-aminolevulinic, bevacizumab, temozolomide, lomustine and carmustine.

- 5-aminolevulinic: better described as an “intraoperative optical imaging agent” vs “diagnostic” or “fluorescent dye”. Also, it received ODD by the FDA in , and FDA approval in 2017. It is not clear what specific regulatory event is being listed as 2002.

- Bevacizumab: This is an anti-VEGF antibody not a radioconjugate small molecule. Also, bev was approved by the FDA in 2009 so again it is not clear the regulatory event being listed as 2014.

- Prolifeprosan 20 with carmustine (GLIADEL) is an implant to deliver the approved drug carmustine. This needs to be clarified.

- Temozolomide: check year of designation date.

Comment 3: We thank the reviewer for pointing out these issues. Indeed, we have now renewed this table, corrected the spelling and content errors, “Target structure” was corrected to “description”. “Year of designation” has now been corrected to “Year of approval” which is a more important category in the context of our paper.

Comment 4: Overuse of “:”. In most contexts it would be more appropriate to replace the “:” and instead start a new sentence. If the authors choose to use a “:” to join 2 related sentences, then the second sentence should not be capitalized.

Answer 4: We have now proof-read the manuscript carefully and changed the “:” to a full stop in many instances which was deemed to be more appropriate.

Comment 5: Review should be carefully proof-read.

Answer 5: The review was carefully proof-read. In addition, the manuscript was checked for language by a native speaker with a PhD degree in science as mentioned in the acknowledgement section.

Reviewer #3: 

Comment 1: The authors present a cross sectional assessment of the US orphan drug development program with regards to glioblastoma. This original research provides a helpful overview of the evolving treatment landscape for GBM, even while highlighting the relative paucity of approved therapies thus far. The statistical analysis in this article is largely descriptive and not problematic. However, the authors' main conclusion that the number of orphan drugs being developed is escalating is not supported by Figure 1, which shows a peak in 2016 and a downward trend since. (2020 results could be affected by COVID-19, but there isn't a clear explanation for 2017-2019.) This doesn't invalidate the analysis, but it should modify the conclusion.

Answer 1: We are very grateful for the overall assessment of our work and have modified/corrected the conclusion that the number of designated drugs is escalating. 

Comment 2: The official WHO term is now "glioblastoma" not glioblastoma multiforme. Clinical outcomes for glioblastoma HAVE improved, just not much.

Answer 2: These two aforementioned points have now been modified in the manuscript and changed accordingly. 

Comment 3: Table 1 is not accurate. Bevacizumab is the angiogenesis inhibitor, BCNU wafers are not.

Answer 3: corrected.

Comment 4: Might be worth noting in the discussion that of these approved therapies, usage of 5-ALA and BCNU wafers is not universal, thus further highlighting the need for more progress.

Answer 4: we have clarified this. 

Comment 5: - Characterizing cilengitide as "well known" and other drugs as "less well-known" seems somewhat arbitrary and akin to a personal opinion.

Answer 5: we have clarified this.

Comment 6: - "Cannaboids" is misspelled.

Answer 6: corrected 

Comment 7: - "Temozolomide" is misspelled.

Answer 7: corrected

Comment 8: - I don't think it's accurate to "assume" that CAR T-cells or dendritic cell vaccines will ultimately be approved for GBM. There is no evidence to support this at this time.

Answer 8: Thank you very much for this important comment, we have clarified this.

Comment 9: - Might be worth discussing that although bevacizumab is still FDA-approved for recurrent GBM, there was considerable discussion about pulling this approval when phase 3 trials demonstrated the absence of a strong survival benefit.

Answer 9: we have added this statement into the discussion section.

Comment 10: The GSK-J4 example doesn't seem to fit this article, as it is not a orphan designated drug.

Answer 10: We thank you very much for this comment, we have amended the text accordingly.

Reviewer #4: 

Comment 1: This manuscript gives a comprehensive overview of the landscape of designated and approved orphan drugs to treat glioblastoma multiforme. The information provided is useful for researchers who work on developing new drugs for GBM or clinicians interested in approved drugs and recent trends. However, the paper has several flaws that need to be corrected (see below). The conclusion that the rate of designated (but not yet approved) drugs per year has steadily increased is not supported by figures 1 and 2, which show a rapid rise in new designated drugs starting about 10 years ago, reaching a maximum in 2016 and then declining to the level of the beginning of the decade. The authors should discuss this trend and potential reasons for the reversal in the Discussion section.

Answer 1: Thank you very much for this important comment. We have corrected the description of the curve’s kinetics accordingly. In addition, we have added a paragraph about potential reasons.

Comment 2: A similar review, entitled "Orphan drugs in glioblastoma multiforme: a review," was published by Lassen et al. in 2014 (https://doi.org/10.2147/ODRR.S46018). The authors should reference that paper and make it clear that their review adds new information.

Answer 2: Thank you very much for pointing out this important paper. We have added it to the references and included a contextual statement into the manuscript’s discussion section as suggested. 

Comment 3: Table 1 has many errors, including typos (ame, copound) and wrong drug information under Target structure and Year of Designation. Please correct these errors.

Answer 3: This is a very well taken point which has also been brought up by reviewer 2. We have now corrected the table (and added lomustine as a still missing approved compound). 

Comment 4: The caption of Figure 1 should be "Barplot shows the number of new orphan drug designations for the indication glioblastoma per year. Years without a new designation are not shown.

The vertical axis label of Figure 1 should be "Number of new drugs per year." Make the same changes to FIgure 2.

Answer 4: corrected

Comment 5: In Figure 3A, it seems that the Lymphoma category does not have a bar.

Answer 5: corrected

Comment 6: In Figure 3A and B, the authors included non-oncological, rare diseases such as Thalassemia and SCD (Sickle Cell Disease). They should carefully check each category and remove all non-oncological classifications.

Answer 6: done

Comment 7: In Figure 4 A, I suggest replacing 'Cellular product/Virus' with 'Cellular/Viral Immunotherapy' for clarity.

Answer 7: this has been changed in Figure 4

Comment 8: In Figure 4 A, the individual plots are 'dot plots' not 'boxplots'; no need to capitalize that phrase.

Answer 8: corrected

Comment 9: In the Abbreviations table, the entries PNH Paroxysmal nightly hemoglobinuria and SCD Sickle cell disease should be removed because they are rare genetic diseases; also see comment about Figure 3 A and B above.

Answer 9: corrected

Comment 10: In the Abbreviations table, replace 'Vasoendothelial growth factor' with 'Vascular endothelial growth factor'.

Answer 10: done

Comment 11: In the Abbreviations table, add VEGFR - Vascular endothelial growth factor receptor.

Answer 11: done

---

## [Decision Letter · Decision Letter 1]

26 May 2021

The drug development pipeline for glioblastoma - a cross sectional assessment of the FDA Orphan Drug Product designation database

PONE-D-21-03939R1

Dear Dr. Johann,

We’re pleased to inform you that your manuscript has been judged scientifically suitable for publication and will be formally accepted for publication once it meets all outstanding technical requirements. Please consider addressing the reviewer's suggestion in your final submission.

Kind regards,

Christopher Wheeler

Academic Editor

PLOS ONE

Additional Editor Comments (optional):

Reviewers' comments:

Reviewer's Responses to Questions

**Comments to the Author**

1. If the authors have adequately addressed your comments raised in a previous round of review and you feel that this manuscript is now acceptable for publication, you may indicate that here to bypass the “Comments to the Author” section, enter your conflict of interest statement in the “Confidential to Editor” section, and submit your "Accept" recommendation.

Reviewer #2: All comments have been addressed

Reviewer #4: All comments have been addressed

2. Is the manuscript technically sound, and do the data support the conclusions?

Reviewer #2: Yes

Reviewer #4: Yes

3. Has the statistical analysis been performed appropriately and rigorously? 

Reviewer #2: Yes

Reviewer #4: Yes

4. Have the authors made all data underlying the findings in their manuscript fully available?

Reviewer #2: Yes

Reviewer #4: Yes

5. Is the manuscript presented in an intelligible fashion and written in standard English?

Reviewer #2: Yes

Reviewer #4: Yes

6. Review Comments to the Author

Reviewer #2: 1. Consider rephrasing abstract and/or Table 1 to be consistent with listing either 4 vs 5 approved therapies for GBM.

2. Consider editing introduction to include more than one example…. “that elaborate combination possibilities between classical therapeutic backbones (i.e., radiotherapy and chemotherapy) and novel,…”

3. It seems more appropriate to replace “genetic lesion” with “genetic mutation” in the context used by the authors.

Reviewer #4: (No Response)

7. PLOS authors have the option to publish the peer review history of their article (what does this mean?). If published, this will include your full peer review and any attached files.

Reviewer #2: No

Reviewer #4: No

---

## [Editor Report · Acceptance letter]

27 May 2021

PONE-D-21-03939R1 

"The drug development pipeline for glioblastoma - a cross sectional assessment of the FDA Orphan Drug Product designation database" 

Dear Dr. Johann:

I'm pleased to inform you that your manuscript has been deemed suitable for publication in PLOS ONE. Congratulations! Your manuscript is now with our production department. 

Kind regards, 

on behalf of

Dr. Christopher Wheeler 

Academic Editor

PLOS ONE